Chronic sleep deprivation is associated with delayed puberty onset in rats, activation of proinflammatory cytokines and gut dysbiosis

Gunawan Shirley Priscilla 1
Huang Shih-Yi 1 2
Hsu Jhih-Wei 3
Lin Chia-Yuan 3 4
Nguyen Nam Nhat 5
Tung Te-Hsuan 2
Liang Shu-Ling 6
Lee Gilbert Aaron 7 8 9 10
Su Chien-Tien ctsu@tmu.edu.tw 11 12
Chen Yang Ching melisa26@tmu.edu.tw 1 2 3 7 8 13 14
1 Graduate Institute of Metabolism and Obesity Sciences, Taipei Medical University , Taipei , Taiwan
2 School of Nutrition and Health Sciences, Taipei Medical University , Taipei , Taiwan
3 Department of Family Medicine, School of Medicine, College of Medicine, Taipei Medical University , Taipei , Taiwan
4 Department of Food Science, National Taiwan Ocean University , Keelung , Taiwan
5 College of Medicine, Taipei Medical University , Taipei , Taiwan
6 Department of Physiology and Pharmacology, College of Medicine, Chang Gung University , Taoyuan , Taiwan
7 Child Development Research Center, Taipei Medical University Hospital , Taipei , Taiwan
8 TMU Research Center for Digestive Medicine, Taipei Medical University , Taipei , Taiwan
9 Department of Microbiology and Immunology, School of Medicine, College of Medicine, Taipei Medical University , Taipei City , Taiwan
10 Department of Medical Research, Taipei Medical University Hospital , Taipei , Taiwan
11 School of Public Health, Taipei Medical University , Taipei , Taiwan
12 Department of Family Medicine, Taipei Medical University Hospital , Taipei , Taiwan
13 Department of Family Medicine, Wan Fang Hospital, Taipei Medical University , Taipei , Taiwan
14 Nutrition Research Center, Taipei Medical University Hospital , Taipei , Taiwan
Brygadyrenko Viktor
Electronic publication date: 2025 Jul 9
Publication date: 2025
Volume: 13
Electronic Location ID: e19668
Received 2024 Dec 6; Accepted 2025 Jun 5
Copyright: ©2025 Gunawan et al.
Copyright year: 2025
Copyright holder: Gunawan et al.
License: This is an open access article distributed under the terms of the Creative Commons Attribution License, which permits unrestricted use, distribution, reproduction and adaptation in any medium and for any purpose provided that it is properly attributed. For attribution, the original author(s), title, publication source (PeerJ) and either DOI or URL of the article must be cited.
License URL: https://creativecommons.org/licenses/by/4.0/

Keywords: Rodent model, Chronic sleep deprivation, Proinflammatory cytokines, Puberty onset, Gut microbiome, Rodent model

Funding: National Science and Technology Council NSTC 112-2314-B-038 -051 -MY3 Taipei Medical University Hospital 110TMU-TMUH-04; 112TMU-TMUH-02-1 Translational Laboratory and the Department of Medical Research at Taipei Medical University Hospital The Taipei Medical University Core Laboratory of Human Microbiome This work was supported by the National Science and Technology Council (NSTC 112-2314-B-038 -051 -MY3) and the Taipei Medical University Hospital (110TMU-TMUH-04; 112TMU-TMUH-02-1). The Translational Laboratory and the Department of Medical Research at Taipei Medical University Hospital supported the preparation of gut microbiome samples. The Taipei Medical University Core Laboratory of Human Microbiome provided technological and analytical support. The funders had no role in study design, data collection and analysis, decision to publish, or preparation of the manuscript.

==============================
Chronic sleep deprivation (CSD) in adolescents has become a trend with adverse health outcomes. Previous studies have demonstrated that sleep deprivation causes inflammation, alters puberty onset, and changes the gut microbiome composition; however, the relationship between these is still unknown. Therefore, we hypothesized that CSD delays the onset of puberty via elevating proinflammatory cytokines and alter ation of gut microbiome composition. Using the modified multiple platform method, we conducted a 4-week CSD experiment in juvenile rats and assessed pubertal markers, antioxidant activity, cytokine levels, and gut microbiome profiles. CSD significantly reduces body weight, delays onset of puberty, and elevated antioxidant enzyme activities in both sexes. In the sleep-deprivation female (SDF) rats, plasma levels of lipopolysaccharide–binding protein (LBP), interleukin-1β (IL-1β), interleukin-6 (IL-6), and tumor necrosis factor-α (TNF-α) were significantly elevated; mRNA levels of TNF-α and IL-1β were also significantly elevated in the colon and reproductive organs, respectively. In the sleep-deprivation male (SDM) rats, only plasma levels of IL-6 were elevated considerably; in addition, mRNA levels of IL-1β and TNF-α were also significantly elevated in the colon and reproductive organs, respectively. Gut microbiome analysis revealed that the predominant bacteria at the genus level were Muribaculaceae, Prevotellaceae UCG-001, and Ruminococcaceae UCG-005 in the SDF rats; Prevotellaceae NK3B31, Ruminococcaceae UCG-010, Eubacterium coprostanoligenes, and Shuttleworthia in the SDM rats. CSD rats with abundant genera were positively correlated with antioxidant enzyme activities and mRNA levels of proinflammatory cytokines. Overall, CSD is associated with delayed puberty onset, possibly via an increase in the expression levels of proinflammatory cytokines and altering the gut microbiome composition, indicating proinflammatory cytokines and gut microbiome play an important role in pubertal timing change. These findings may guide the future studies to intervene sleep deprivation-related delays in the onset of puberty.

INTRODUCTION

The definition of puberty is a phase of human development where rapid growth and the ability to reproduce begins (Wood, Lane & Cheetham, 2019). The early onset of puberty has become a secular trend in adolescents worldwide (Hardy et al., 2006; Hui et al., 2012). Adolescence is a phase of life that spans between childhood and adulthood (Sawyer et al., 2018), and it has been characterized by decreased sleeping patterns over time (Matricciani, Olds & Petkov, 2012; Shochat, Cohen-Zion & Tzischinsky, 2014). Studies have revealed an association between sleep duration and the onset of puberty in adolescents (Hoyt et al., 2018; Sadeh et al., 2009; Wang et al., 2020). Chronic sleep deprivation (CSD) leads to the accumulation of reactive oxygen species (ROS), which are natural byproducts of cellular metabolism and play important roles in cell signalling. However, excessive ROS generation, such as during CSD, can lead to oxidative stress—an imbalance between pro-oxidants and antioxidants in the body. To counteract this, the body activates endogenous antioxidant defense systems, including enzymes such as superoxide dismutase (SOD), catalase (CAT), and glutathione peroxidase (GPx). These enzymes neutralize ROS, prevent cellular damage, and restore redox balance. This adaptive response is part of a tightly regulated feedback mechanism aimed at maintaining cellular homeostasis and preventing oxidative injury to lipids, proteins, and DNA (Birben et al., 2012).

Major proinflammatory cytokines, such as interleukin-1β (IL-1β), interleukin-6 (IL-6), and tumor necrosis factor-alpha (TNF-α), are also associated with sleep deprivation (Garbarino et al., 2021; Mullington et al., 2010). Sleep deprivation leads to an inflammatory response, especially in the hypothalamus, inhibiting gonadotropin-releasing hormone (GnRH) expression, which reduces luteinizing hormone (LH) release and may alter the onset of puberty (Haziak et al., 2018).

In addition, some studies have shown that sleep deprivation alters the composition of the gut microbiome (Poroyko et al., 2016; Reynolds et al., 2017). The recent study by Poroyko et al. (2016) in mice suggests that chronically disturbed sleep is associated with increased food intake, disruption of gut microbiota. Reynolds et al. (2017) reported that shift work could produce dysbiosis. Benedict et al. (2016) also showed the relationship between sleep disturbance and gut microbiota in young individuals as participants exhibited a significant increase in Firmicutes–Bacteroides ratio. On the other hand, the association between gut microbiome and puberty onset has not been well established; one study observed that the composition of the gut microbiome is different between pubertal and non-pubertal groups (Yuan et al., 2020). Dong et al. (2019) elucidated gut microbiota discrepancy between early puberty and healthy participants. Similarly, Huang et al. (2022) found distinct gut microbiota compositions and functions between different subphenotypes of PP and healthy populations. Another animal study also demonstrated that probiotics can reverse the early onset of puberty in rats (Cowan & Richardson, 2019). Thus, alteration of gut microbiome composition might influence the onset of puberty.

We hypothesized that CSD delays the onset of puberty via elevating inflammation (as assessed by inflammatory markers) and alteration of gut microbiome composition. Here, we performed a sleep deprivation study using animal models to establish the relationship between proinflammatory cytokines, gut microbiome, and pubertal timing.

Materials and Methods

Animal study

We conducted this study to investigate the impact of sleep deprivation on altering the puberty onset in juvenile Sprague-Dawley (SD) rats. Four pregnant SD rats were purchased from BioLASCO Taiwan Co. Ltd and housed at the Laboratory Animal Center at Taipei Medical University in a controlled environment (12-hour light-dark cycle, 22–24 °C, 40%–60% humidity). The juvenile SD rats were weaned and grouped at postnatal day 21 (PND 21). Then, they were randomly assigned to control female (CF) (n = 6), sleep-deprivation female (SDF) (n = 6), control male (CM) (n = 6), and sleep-deprivation male (SDM) (n = 6) groups, a total of 24 rats. We ensured their body weight and sex were balanced across groups. They were subjected to 15 h of sleep deprivation per day for 4 weeks after weaning. Sleep deprivation is a highly stressful condition; therefore, the body weights of rats were monitored every other day, and all rats survived until euthanization. The rats were euthanized by using cardiac puncture after 4 weeks of sleep deprivation, and blood samples and tissues were collected and stored in the −80 °C refrigerator until used.

CSD model

The rats in the SDF and SDM groups were subjected to sleep deprivation for 15 h (from 08:30 to 23:30) per day for 4 weeks continuously based on the modified inverted flowerpot method (Machado et al., 2004). Sleep deprivation for 4 weeks was considered as CSD (Poroyko et al., 2016; Zhu et al., 2015). The SD rats were placed inside a water tank containing eight circular platforms, each with a diameter of six cm. They were put in the tanks with the same sex and group-housed after they returned to the home cage. Water was added to within one cm of the upper surface of each platform. Each water tank had at most six rats. When the rats reached the rapid eye movement stage of sleep, muscle atonia caused them to fall into the water, at which point they had to climb up a platform to avoid being drowned. The control rats were housed in standard cages in the same animal room under identical light-dark cycles, temperature, humidity, and handling frequency. However, they were not placed in water tanks or exposed to the inverted flowerpot environment, which may have resulted in different levels of physical or psychological stress compared to the SD groups. This difference in environmental exposure is acknowledged as a limitation of the study.

Assessment of the onset of puberty

The onset of puberty in the female rats was started observed at PND25 and determined by using vaginal smears. Vaginal smears were performed by inserting a sterilized pipette tip filled with 10 µL of normal saline into the vagina and then taking the fluid. The fluid was placed in the glass slide and observed under microscope vision. Preestrous, estrous, metertrous, and diestrous of estrous cycle stages were identified by cell morphology (Marcondes, Bianchi & Tanno, 2002). In addition, the onset of puberty in the male rats was started at PND 35 and determined by preputial separation (Korenbrot, Huhtaniemi & Weiner, 1977). We pre-specified puberty-onset timing as our primary hypothesis; all other comparisons are exploratory and unadjusted for multiplicity.

Tissue preparation

Colon (cecum), ovary, and testes samples were dissected and washed immediately in 0.1 M phosphate buffer saline (PBS). They were homogenized on ice in PBS 1:2 (w/v; one g tissue with three mL PBS, pH 7.4) and centrifuged at 10,000× g for 15 min at 4 °C. The supernatants were collected to determine CAT, SOD, and glutathione GPx activity.

Protein determination

The protein levels of the colon, ovary, and testes homogenates were determined by using the Bradford method (Bradford, 1976).

Antioxidant enzyme activities

Antioxidant enzyme activities were determined by using CAT, SOD, and GPx activities. Antioxidant enzyme activities were analyzed by using assay kits from Cayman Chemical (Ann Arbor, MI, USA). Analyzed procedures followed the manufacturer’s protocols: CAT (item no. 707002), SOD (item no. 706002), and GPx (item no. 703102). Antioxidant enzyme activities were normalized to the total protein in the homogenates and expressed as units per mg of protein.

Circulating lipopolysaccharide-binding protein (LBP) levels and proinflammatory cytokines (IL-1β, IL-6, and TNF-α) were quantified in rat plasma samples. Plasma LBP levels were measured using an enzyme-linked immunosorbent assay (ELISA) kit (Cusabio; CSB-E11184r). Briefly, 100 µL of plasma samples and standards were added to a 96-well plate pre-coated with LBP-specific antibodies and incubated for 2 h at 37 °C. After washing, an HRP-conjugated secondary antibody was added and incubated for 1 h. TMB substrate was then applied, and the color reaction was stopped with a stop solution. Absorbance was measured at 450 nm using a microplate reader.

IL-1β, IL-6, and TNF-α levels were assessed using the LEGENDplex™ Multi-Analyte Flow Assay Kit (Cat. 741395 and 741396; BioLegend, San Diego, CA, USA) based on flow cytometry. In brief, 25 µL of each plasma sample was incubated with a mixture of capture beads coated with cytokine-specific antibodies. After a 2-hour incubation at room temperature, biotinylated detection antibodies were added, followed by streptavidin-PE. Samples were analyzed on a flow cytometer, and cytokine concentrations were calculated using standard curves generated with LEGENDplex™ Data Analysis Software Suite (BioLegend, San Diego, CA, USA).

RNA extraction and real-time quantitative reverse transcription-polymerase chain reaction

The RNA was extracted by using the RNeasy Mini Kit (Qiagen, Hilden, Germany) in accordance with the manufacturer’s protocol. Following RNA isolation, one µg of RNA was used for reverse transcription to cDNA by using the MMLV Reverse Transcription Kit (Protech Technology Enterprise, Co., Ltd., Taipei, Taiwan). cDNA was used to quantify the transcript levels on the Smart Quant Green Master Mix system (Protech Technology Enterprise, Co., Ltd., Taipei, Taiwan). GAPDH is used as an internal control (Ct of target gene−Ct of GAPDH = ΔCt), and ΔCt of control is used as the calibrator (ΔCt of sample−ΔCt of calibrator = ΔΔCt). Relative mRNA levels of target genes = 2−ΔΔCT (fold change vs. control).

DNA sequences of rat-specific primers were summarized as follows:

Gene symbol	Forward primer 5′ to 3′(F)	Reverse primer 5′ to 3′(R)	
GAPDH	5′-GTGCCAGCCTCGTCTCATAG-3′	5′-CGTTGATGGCAACAATGTCCA-3′	
TNF-α	5′- CTCTTCTCATTCCTGCTCGT-3′	5′-GGGAGCCCATTTGGGAACTT-3′	
IL-1B	5′-CACCTCTCAAGCAGAGCACA-3′	5′-TCCTGGGGAAGGCATTAGGA-3′	
IL-6	5′-ACCCCAACTTCCAATGCTCT-3′	5′-AGCACACTAGGTTTGCCGAG-3′	

Gut microbiome composition

Fecal samples were collected at vaginal opening days (PND 30∼PND 40) from female rats and were collected at preputial separation days (PND 39∼PND 49) from male rats during the experiment. In brief, collected fecal samples were transferred immediately to cold storage and remained stored at −80 °C until processing near days of vaginal opening. Fecal genomic DNA was extracted using the QIAamp DNA Stool Mini Kit (cat. no. 51504, Qiagen, Hilden, Germany) according to the manufacturer’s instructions, stored at −80 °C, and underwent processing including polymerase chain reaction (PCR) assays and 16S rRNA sequencing. The PacBio sequencing for full-length 16S genes (V1–V9 regions) was performed. The full-length 16S genes was amplified using barcoded 16S gene-specific primers. Subsequently, the PCR reaction was carried out by KAPA HiFi HotStart ReadyMix (Roche, Basel, Switzerland), and its products were purified using the AMPure PB Beads for SMRTbell library construction and sequencing processes. Consequently, multiple sequence alignment was performed by QIIME2 alignment MAFFT against the NCBI database to analyze the sequence similarities among the amplicon sequence variants (ASVs).

Operational taxonomic unit (OTU) clustering and taxonomic analysis were performed using Genomics workbench v.22.0 (CLC Bio, Denmark). The sequences were trimmed, merged, and clustered into OTUs at 97% sequence similarity based on the SILVA v.32 database using CLC Microbial Genomics Module. Within sample microbial diversity (alpha diversity) were assessed using the phyloseq package in R from normalized operational taxonomic unit (OUT) abundances. Between group diversity (beta diversity) was calculated by evaluating the unique taxa present in each group relative to the shared taxa across both. We used the principal coordinate analysis (PCoA) to explore the inter-sample variation, grounded in Bray–Curtis dissimilarity. For unsupervised grouping of the samples, hierarchical clustering was executed employing Bray–Curtis distance with a complete linkage method. LEfSe analysis was performed to detect bacterial taxa with significantly different abundance between the control and sleep deprivation groups; significance was indicated if the linear discriminant analysis value was >2.0 with p < 0.05.

Statistical analysis

Values are presented as mean ± standard error of the mean, and all data were examined the normality by using Shapiro–Wilk test. A student’s t-test was performed to compare two groups. ANOVA was performed for three groups, following Tukey’s multiple comparisons test via GraphPad Prism 8.0.1 software. Heatmaps were plotted via R version 4.0.3 (R Foundation for Statistical Computing, Vienna, Austria), which showed Spearman’s rank correlation coefficient between the abundance of bacterial taxa and gene/protein levels. Differences were considered significant at p < 0.05. Data processing and visualization were performed by using the heatmap package (version 1.0.12), along with phyloseq (1.34.0), ggplot2 (3.3.3), and dplyr (1.0.2). Correlation analyses were based on Spearman’s rank correlation using the cor.test() function in base R, and microbiome data structures were managed using the phyloseq package.

Ethical approval

The Taipei Medical University Institutional Animal Care and Use Committee (IACUC/IACUP) approved all animal procedures (approval no. LAC-2020-0048). All procedures were conducted in accordance with the Taiwan Code of Practice for the care and use of animals for scientific purposes.

Results

CSD attenuated growth status and delayed onset of puberty in juvenile rats

Body weight and pubertal timing in rats were monitored during experiment. The results revealed that CSD for 4 weeks is associated with a decrease in body weight in female (SDF:151.90 ± 3.84 g vs. CF:168.6 ± 3.33 g) and male rats (SDM: 197.30 ± 2.72 g vs. CM: 218.90 ± 4.82 g) (Fig. 1A); in addition, pubertal timing also delayed in female rats (SDF: 36.33 ± 1.17 day vs. CF: 31.00 ± 0.89 day) and male rats (SDM: 45.17 ± 0.65 vs. CM: 40.33 ± 0.76 day) (Fig. 1B). Biochemical characteristics and organs/tissues weight analysis revealed that CSD significantly attenuated blood levels of total protein and albumin in female and male rats but significantly increased levels of triglyceride and aspartate aminotransferase (AST) in male rats (Table S1); in addition, CSD significantly attenuated weights of muscle in female rats and significantly attenuated brain, muscle epididymal white adipose tissue, liver, kidney, seminal vesicle, and epididymis in male rats (Table S2). Taken together, CSD for 4 weeks significantly attenuates growth status and delays the onset of puberty in both female and male rats. In our model, this delayed pubertal timing may be partly attributed to reduced somatic growth, systemic inflammation, and oxidative stress—factors known to influence the hypothalamic-pituitary-gonadal (HPG) axis.

Figure 1 Sleep deprivation decreases body weight and delays the onset of puberty in both sexes.

Sleep deprivation-treated groups consistently have lower body weight compared to control groups. Sleep deprivation delays vaginal opening and preputial separation in female and male rats, respectively. Dots represent group means ± SEM (n = 6 rats/group). *p < 0.05; **p < 0.01; ***p < 0.001.

Figure 2 Sleep deprivation induces elevated antioxidant enzyme activity.

(A) The comparison of antioxidant enzyme activities in the colon of rats in the control and sleep deprivation groups is shown. (B) The comparison of antioxidant enzyme activities in reproductive organs (female: ovary, male: testis) of rats in the control and sleep deprivation groups is shown.Data are presented in as mean ± SEM (n = 6 rats/group). *p < 0.05; **p < 0.01; ***p < 0.001.

CSD increases antioxidant enzyme activities in reproductive organs in female and male rats

To determine the effects of CSD on antioxidant responses, we investigated the antioxidant enzyme activities in the colon and reproductive organs. In the colon organ, we observed that the antioxidant enzymes as CAT, SOD, and GPx activities were not significantly changed between CF and SDF rats (Fig. 2A, superior section); however, CAT (102.50 ± 15.56 nmol/mg protein/min), SOD (22.27 ± 2.70 U/mg protein), and GPx (158.20 ± 10.31 nmol/mg protein/min) in SDM rats were significantly higher than those of CM rats (Fig. 2A, inferior section). In reproductive organs, we observed that CAT (20.66 ± 1.63 nmol/mg protein/min) and SOD (1.18 ± 0.05 U/mg protein) activities in SDF rats were significantly higher than those of CF rats (Fig. 2B, superior section); in addition, CAT (13.75 ± 0.77 nmol/mg protein/min), SOD (0.925 ± 0.03 U/mg protein) and GPx activities (114.4 ± 3.88 nmol/mg protein/min) in SDM rats were significantly higher than those of CM rats (Fig. 2B, inferior section). Taken together, CSD is associated with an increase in antioxidant enzyme activities in reproductive organs of both sexes; however, antioxidant enzyme activities in colon organs have only increased in male rats.

CSD leads to inflammation in the colon, reproductive organs, and circulatory system in female and male rats

Next, we investigated whether CSD is leading to an inflammatory response in female and male rats’ colon, reproductive organs, and circulatory systems. Therefore, we determined protein levels of lipopolysaccharide-binding protein (LBP), IL-1β, IL-6, and TNF-α in the circulatory system and mRNA levels of those in the colon and reproductive organs.

The results revealed that the protein levels of LPS (2,424 ± 153.2 pg/ml), IL1-β (0.132 ± 0.048 pg/ml), IL-6 (10.64 ± 2.85 pg/ml), and TNF-α (1.06 ± 0.11 g/ml) in the plasma of SDF rats were significantly higher than those of CF rats (Fig. 3A, superior section). The protein levels of IL-6 (13.32 ± 2.93 pg/ml) in SDM rats’ plasma were significantly higher than those of CM rats (Fig. 3A, inferior section). In the colon, mRNA levels of TNF-α in SDF rats were considerably higher than those of CF rats (2.86 ± 0.33 old vs. CF); mRNA levels of IL-1β in SDM rats were significantly higher than those of CM rats (0.98 ± 0.24 fold vs. CF) (Fig. 3B). In reproductive organs, mRNA levels of IL-1β in SDF rats were significantly higher than those of CF rats (1.62 ± 0.17 fold vs. CF); mRNA levels of TNF-α in SDM rats were significantly higher than those of CM rats (1.26 ± 0.07 fold vs. CM) (Fig. 3C). CSD leads to inflammation in the colon, reproductive organs and circulatory system, especially in female rats.

Figure 3 Sleep deprivation is associated with an increase in LBP and proinflammatory cytokines levels in circulation, colon, and reproductive organs in female and male rats.

(A) Comparison of LBP, IL-1β, IL-6, and TNF-α protein levels in the circulatory system between control and sleep-deprived groups in female (superior section) and male (inferior section) rats. (B) Comparison of mRNA levels of IL-1β, IL-6, and TNF-α in the colon between control and sleep-deprived groups in female (left section) and male (right section). (C) Comparison of TNF-α, IL-1β, and IL-6 mRNA levels in reproductive organs between control and sleep-deprived groups in female (left section) and male (right section) rats. Data are presented as mean ± SEM (n = 6 in each group), *p < 0.05; **p < 0.01; ***p < 0.001.

CSD alters gut microbiome composition

Next, we investigated the association between CSD and the composition of the gut microbiome. First, α-diversity (including Shannon and Simpson index) analysis indicated the sleep deprivation groups were significantly lower than control groups in both sexes (Fig. S1). Second, β diversity analysis indicated the distinct clustering of the microbiome compositions between the control and the sleep-deprived groups, and the result revealed a significant difference between CF and SDF groups (p = 0.013), as well as CM and SDM groups (p = 0.006) (Fig. 4A). Furthermore, the relations between specific bacterial taxa and sleep deprivation in both sexes were determined by using LEfSe analysis. The predominant bacteria at the genus level were Muribaculaceae, Prevotellaceae UCG-001, and Ruminococcaceae UCG-005 in the SDF group, and Prevotellaceae NK3B31, Ruminococcaceae UCG-010, Eubacterium coprostanoligenes, and Shuttleworthia in the SDM group (Fig. 4B).

Figure 4 Sleep deprivation alters gut microbiota composition in the sleep deprivation-treated rat groups.

(A) The comparison of β-diversity patterns of rats in the control and sleep deprivation groups is shown. (B) The comparison of abundant bacterial taxa of rats in the control and sleep deprivation groups is shown. Different colors of linear discriminant analysis effect size (LEfSe) indicate the group in which clade was most abundant (n = 6 rats/group). Significant bacterial genera were determined by Kruskal-Wallis test (p < 0.05) with LDA score greater than 2.

Correlation among abundant genera, pubertal timing, antioxidant enzyme activity, and inflammatory cytokines

Abundant genera were involved in pubertal timing in female and male rats, and the results revealed that g_Ruminococcaceae_UCG-005 was positively correlated with vaginal opening day (Spearman’s rho = 0.633, p = 0.027) in SDF group, whereas g_Roseburia was negatively correlated with vaginal opening day (Spearman’s rho = −0.696, p = 0.012) in CF group (Table S3). In addition, g_Prevotellaceae_NK3B31_group was positively correlated with preputial separation day (Spearman’s rho = 0.587, p = 0.045) in SDM group, whereas g_Lachnospiraceae_A2, g_Ruminiclostridium_9, g_Clostridium_sensu_stricto_1 and g_Clostridiales_vadinBB60_group _Uncultured were negatively correlated with preputial separation day in CM group (Table S4).

The heat maps showed the correlations between abundant genera and antioxidant enzyme activity (Figs. 5A and 5B) and the correlations between abundant genera and inflammation (Figs. 5C and 5D). The results revealed the abundant genera in the SDF and SDM groups were positively correlated with antioxidant enzyme activity and inflammation; in contrast, the abundant genera in the CF and CM groups were negatively correlated with antioxidant enzyme activity and inflammation.

Figure 5 Abundant bacterial taxa in sleep deprivation rats were associated with elevated antioxidant enzyme activities and proinflammatory mRNA levels.

(A–B) Heat map depicting associations between abundant bacterial taxa and antioxidant enzyme activities of rats in the control and sleep deprivation groups is shown. (C–D) Heat map depicting associations between abundant bacterial taxa and proinflammatory mRNA expression levels of rats in the control and sleep deprivation groups is shown. p value is determined with Spearman’s correlation; *p < 0.05.

Discussion

The present study’s findings suggest that CSD is associated with delayed puberty onset, and attenuated body weight in juvenile rats. Moreover, we observed inflammation and gut microbial taxonomy alterations in the colon, subsequently affecting reproductive organs.

Previous animal studies have shown that sleep-deprived rats have lower body weight than control rats (Everson & Szabo, 2011; Koban et al., 2008; Lai et al., 2022), and our study indicated likewise. For example, Everson & Szabo (2011) subjected adult male rats to repeated cycles of total sleep deprivation over several weeks, resulting in significant weight loss and metabolic dysregulation. Similarly, Koban et al. (2008) used the inverted flowerpot method in young adult male rats and found that chronic REM sleep deprivation caused sustained hyperphagia without compensatory weight gain. Our findings align with Lai et al. (2022), who reported that chronic sleep-deprived adolescent rats of both sexes (PND 21 onward) exhibited growth suppression and altered gut microbiota. In contrast, several human epidemiological studies (Schmid et al., 2008; Taheri et al., 2004) have found that shorter sleep duration correlates with weight gain, likely due to compensatory increases in caloric intake, hormonal changes (e.g., ghrelin, leptin), and sedentary behavior. These discrepancies may stem from species differences, experimental design, and the nature of stress exposure in animal models, which may suppress appetite or energy metabolism differently than in free-living humans (Deng et al., 2018; Liu et al., 2021; Seo et al., 2020).

In this study, we observed that chronic sleep deprivation (CSD) significantly increased antioxidant enzyme activity in reproductive organs of both sexes (CAT, SOD, and GPx in SDM testes; CAT and SOD in SDF ovaries). Additionally, antioxidant activity increased in the colon of SDM rats, but not in SDF rats. These findings suggest that CSD induces a moderate oxidative stress response that is sufficient to activate compensatory antioxidant defense mechanisms without overwhelming them. The literature on oxidative stress and sleep deprivation presents mixed findings. For instance, Lungato et al. (2013) reported increased SOD activity in splenocytes following sleep deprivation, consistent with our observation of elevated antioxidant enzymes, particularly in males. In contrast, Gao et al. (2019) found decreased antioxidant enzyme activity in intestinal tissues of sleep-deprived mice, which they attributed to severe and prolonged oxidative damage that ultimately suppresses antioxidant defense. Villafuerte et al. (2015) further proposed that chronic oxidative burden may lead to enzyme exhaustion or structural damage, resulting in lower activity levels over time (Lungato et al., 2013). The key difference in our study may lie in the duration and intensity of oxidative stress. Our CSD protocol (15 h/day for 4 weeks) likely induced a moderate level of ROS, triggering upregulation of CAT, SOD, and GPx expression as an adaptive response—without yet reaching the threshold where oxidative damage inhibits enzyme function. This pattern aligns with early or mid-phase oxidative responses, where the redox system remains functionally intact.

We discovered that CSD is associated with an increase in antioxidant enzyme activities in reproductive organs of both sexes; however, in the colon, significant increases in antioxidant enzyme activities were observed only in male rats. The sex-specific response may reflect differences in tissue susceptibility to oxidative stress and immune activation. Previous studies (Maltz et al., 2024; Tan et al., 2023) have shown that male rodents tend to exhibit a more pronounced systemic and intestinal inflammatory response to stress, potentially due to lower basal levels of estrogen, which has known antioxidant and anti-inflammatory properties. As such, male rats may experience greater oxidative stress in the gut under CSD conditions, triggering a compensatory upregulation of antioxidant enzymes such as CAT, SOD, and GPx. In contrast, females may be partially protected by estrogen-mediated pathways, which could explain the absence of a significant antioxidant response in the colon despite systemic inflammation.

In addition, we observed elevated circulating LBP and proinflammatory cytokines (including IL-1β, IL-6, and TNF-α) levels (Fig. 4A). LPS, a major component of the gram-negative bacteria outer membrane, is known as endotoxin, which causes endotoxemia when it is released into the bloodstream (Gnauck, Lentle & Kruger, 2016; Meng et al., 2021). It binds to LBP, which eventually results in the production of cytokines and other proinflammatory mediators (Guha & Mackman, 2001; Meng et al., 2021); therefore, LBP may regulate IL-1β, IL-6 and TNF-α. In addition, the mRNA levels of TNF-α and IL-1B significantly increase in the colon of the SDF and SDM groups, respectively (Fig. 4B); the mRNA levels of IL1-B and TNF-α also significantly increase in the reproductive organs of the SDF and SDM groups, respectively (Fig. 4C). Previous studies have shown that sleep deprivation causes inflammation (Lai et al., 2022; Mullington et al., 2010), these results are consistent with ours. Some ex vivo studies also revealed that production of TNF-α, IL-1β, and IL-6 induced by LPS increases during sleep deprivation (Garbarino et al., 2021). A previous study showed that inflammation in the hypothalamus upregulates the IL-1B gene expression in the hypothalamus, reduces GnRH mRNA levels, and as a consequence, reduces LH release (Haziak et al., 2018); this result supports our findings on how inflammation in sleep-deprived rats is associated with a delay in the onset of puberty, especially in female rats. Inflammation has many pathways and checkpoints; hence, how the results can explain the impact of sleep deprivation and the development of inflammation warrants further investigation.

After the rats had undergone 4 weeks of sleep deprivation, the richness and diversity of their gut microbiomes were significantly decreased regardless of sex. Previous literature has demonstrated that CSD influences gut microbiome composition in both human (Benedict et al., 2016) and animal studies (Poroyko et al., 2016). One human study revealed that sleep disturbance for two days causes significant increase in Firmicutes-Bacterioides ratio (positive correlation with obesity), higher abundances of the families Coriobacteriaceae and Erysipelotrichaceae, and lower abundance of Tenericutes in young individuals (Benedict et al., 2016); in addition, another animal study indicated sleep disruption for four weeks causes an increase of food intake, visceral white adipose tissue and systemic inflammation via changes in gut microbiomes (characterized by the preferential growth Lachnospiraceae and Ruminococcaceae and a decrease of Lactobacillaceae families) (Poroyko et al., 2016). Therefore, CSD causes white adipose tissue or systemic inflammation via gut microbiome dysbiosis; these results are consistent with ours.

Previous studies have shown that f_Muribaculaceae attenuates obesity and is related to body weight loss (Hou et al., 2020; Lagkouvardos et al., 2019). At the genus level, we observed an increased abundance of Prevotellaceae UCG-001, which is positively correlated with the AMPK (AMP-activated protein kinase) activation signaling pathway (Song et al., 2019), and Ruminoccocaceae UCG-005, which has been reported to alleviate obesity (Zhang et al., 2019; Zhao et al., 2017). Thus, the increased abundance of these bacterial taxa may explain reduced body weight in the SDF group. Additionally, a higher abundance of g_Shuttleworthia in the SDM group was found to be correlated with inflammation (Du et al., 2022; Li et al., 2022). As a result, our findings regarding the abundance of bacterial taxa, increased levels of antioxidant enzyme activity, and proinflammatory markers after 4 weeks of sleep deprivation are consistent with those of other studies.

The relationship between microbial dysbiosis and inflammation is multifaceted, involving complex host-microbe and microbe-microbe interactions. In our study, CSD significantly reduced gut microbial alpha diversity, suggesting a loss of microbial richness and evenness—an ecological hallmark of dysbiosis. Reduced diversity has been associated with lower resilience to environmental stress and a shift toward potentially proinflammatory microbial communities (Zheng, Liwinski & Elinav, 2020). Furthermore, beta diversity analysis revealed distinct clustering between control and CSD groups, indicating significant alterations in microbial community composition. These changes were marked by the expansion of specific bacterial genera (e.g., Prevotellaceae UCG-001, Ruminococcaceae UCG-005, Shuttleworthia) that have been implicated in gut permeability, lipopolysaccharide (LPS) production, and systemic inflammation. For instance, an increased abundance of Shuttleworthia has been correlated with elevated serum TNF-α and gut epithelial disruption in rodent models (Du et al., 2022; Li et al., 2022), consistent with our observation of elevated proinflammatory cytokines. From a microbial ecology perspective, CSD appears to create a selective environment favoring taxa with inflammatory potential, possibly due to increased luminal stress, nutrient shifts, or host immune activation. This community shift may promote the release of microbial-associated molecular patterns (MAMPs) like LPS into the circulation, activating Toll-like receptor (TLR) pathways and driving systemic inflammation. Such inflammatory signaling can, in turn, impair neuroendocrine regulation—particularly the hypothalamic–pituitary–gonadal (HPG) axis—and may contribute to the delayed onset of puberty observed in our model. Altogether, our findings suggest that CSD-induced microbial dysbiosis leads to a proinflammatory microbial ecology that compromises gut–immune homeostasis (Thaiss Christoph et al., 2014), ultimately influencing systemic physiology and developmental timing in the host.

Interestingly, while our study and other rodent models suggest that CSD delays the onset of puberty, several human epidemiological studies have reported an association between shorter sleep duration and earlier puberty onset, particularly in girls (Hoyt et al., 2018; Wang et al., 2020). This apparent discrepancy may be attributed to species-specific differences in neuroendocrine regulation, stress response, and environmental confounders. In rodent models, sleep deprivation is often implemented using stressful paradigms (e.g., the inverted flowerpot method), which can elevate corticosterone levels and suppress the hypothalamic-pituitary–gonadal (HPG) axis, thus delaying pubertal development. In contrast, in humans, insufficient sleep is often accompanied by increased screen time, higher adiposity, poor diet, and reduced melatonin secretion, all of which may independently stimulate early HPG axis activation and, thus, earlier puberty. Additionally, psychosocial stressors in humans (e.g., family disruption or urban environments) have been associated with early maturation, potentially overriding the suppressive effects of poor sleep alone. These distinctions underscore the importance of considering the broader physiological and environmental context when interpreting sleep–puberty interactions across species.

A limitation of our study is that we did not obtain data on hormone levels, and kisspeptin, or GnRH neuron expression levels of the hypothalamus at the onset of puberty. Unlike researchers in previous studies, we did not use electroencephalography (EEG) to measure sleep deprivation (Huber, Deboer & Tobler, 2000; Mohammed et al., 2011); however, results from our study and results from other studies appear to be consistent and independent of the use of EEG (Barf et al., 2012; Koban et al., 2008; Mohammed et al., 2011). Ours is the first study to investigate the association between CSD and the onset of puberty in juvenile rats. Another limitation of this study is the lack of environmental control for stress exposure. While rats in the CSD groups were subjected to the inverted flowerpot method—a known physical and psychological stressor—the control rats remained in their home cages and were not exposed to similar handling or water tank conditions. As a result, differences in stress levels between groups could have influenced our findings, independent of sleep deprivation itself. Corticosterone, the primary glucocorticoid in rodents, is a well-established biomarker of stress. Unfortunately, corticosterone levels were not measured in this study, which limits our ability to isolate the effects of sleep deprivation from those of general stress. Previous research has shown that elevated corticosterone can delay pubertal onset, suppress growth via hypothalamic-pituitary axis inhibition, and alter gut microbiota composition by affecting intestinal permeability and immune responses (Bailey et al., 2011). Thus, it is possible that the observed delays in pubertal timing, inflammatory markers, and microbial shifts were partly mediated by stress-related hormonal changes.

In summary, we demonstrated that CSD increases antioxidant enzyme activity (CAT, SOD, and GPx) and inflammation (LBP, IL-1β, IL-6 and TNF-α) as well as alternation of gut microbiome (Prevotellaceae, Ruminococcaceae, Shuttleworthia) plays an important role in antioxidant enzyme activity and inflammation. Future studies are suggested to use prebiotics/postbiotic intervention to change the gut microbiome or treat with some compounds to reduce proinflammatory cytokines to reverse sleep deprivation-related delay in the onset of puberty.

CONCLUSIONS

Our findings suggest that CSD is associated with delayed puberty onset and altered growth profiles in juvenile rats. These physiological changes coincided with increased antioxidant enzyme activities, elevated levels of proinflammatory cytokines (LBP, IL-1β, IL-6 and TNF-α), and significant shifts in gut microbiome composition. While these correlations support a potential mechanistic link between CSD, inflammation, and gut microbial dysbiosis, the data do not establish a direct causal pathway. Further mechanistic studies, particularly involving gut microbiome modulation or cytokine inhibition, are needed to confirm whether these pathways mediate the effects of CSD on pubertal timing. Future interventional or mechanistic studies in the context of sleep-deprivation should focus on treating oxidative stress and gut dysbiosis.

Supplemental Information

Supplemental Information 1 Basic biochemical characteristics

Data are reported as mean ± SEM (female, n = 6 rats/group; male, n = 6 rats/group ), and values with different superscripts are significantly different compared to the control group (p < 0.05). Abbreviations: TP, Total Protein; AST(GOT), aspartate aminotransferase (glutamic-oxaloacetic transaminase); ALT(GPT), alanine transaminase (glutamic pyruvic transaminase); ALKP, alkaline phosphatase, BUN, blood urea nitrogen; CRE, creatinine; TG, triglycerides; CHOL, cholesterol; HDL, high-density lipoprotein; LDL, low-density lipoprotein; PND, post-natal day; CF, control female; SDF, sleep deprivation female; CM, control male; SDM, sleep deprivation male.

Supplemental Information 2 Retrieved organ weight

Data are reported as mean ± SEM (female, n = 6 rats/group; male, n = 6 rats/group), and values with different superscripts are significantly different compared to the control group (p < 0.05) . Abbreviations: PRAT, perirenal adipose tissue; eWAT, epididymal white adipose tissue; CF, control female; SDF, sleep deprivation female; CM, control male; SDM, sleep deprivation male.

Supplemental Information 3 The relationship between abundant bacterial taxa in SDF and CF groups and vaginal opening day

a: p-value is determined by Spearman’s correlation analysis ; * p < 0.05. Abbreviations: CF, control female; SDF, sleep deprivation female.

Supplemental Information 4 The relationship between abundant bacterial taxa in SDM and CM groups and preputial separation day

a: p value is determined with Spearman’s correlation analysis ; * p < 0.05; ** p < 0.01. Abbreviations: CM, control male; SDM, sleep deprivation male.

Supplemental Information 5 α diversity of gut microbiota was analysed in female rat (A) and male rat (B) groups

Abbreviations: CF, control female; SDF, sleep deprivation female; CM, control male; SDM, sleep deprivation male.

Supplemental Information 6 Raw data for Fig. 1

Supplemental Information 7 Raw data for Fig. 2

Supplemental Information 8 Raw data for Fig. 3

Additional Information and Declarations

Competing Interests

Author Contributions

Animal Ethics

Data Availability

The authors declare there are no competing interests.

Shirley Priscilla Gunawan conceived and designed the experiments, analyzed the data, authored or reviewed drafts of the article, and approved the final draft.

Shih-Yi Huang conceived and designed the experiments, performed the experiments, analyzed the data, prepared figures and/or tables, authored or reviewed drafts of the article, and approved the final draft.

Jhih-Wei Hsu analyzed the data, prepared figures and/or tables, authored or reviewed drafts of the article, and approved the final draft.

Chia-Yuan Lin analyzed the data, authored or reviewed drafts of the article, and approved the final draft.

Nam Nhat Nguyen analyzed the data, prepared figures and/or tables, authored or reviewed drafts of the article, and approved the final draft.

Te-Hsuan Tung performed the experiments, analyzed the data, authored or reviewed drafts of the article, and approved the final draft.

Shu-Ling Liang performed the experiments, authored or reviewed drafts of the article, and approved the final draft.

Gilbert Aaron Lee performed the experiments, authored or reviewed drafts of the article, and approved the final draft.

Chien-Tien Su analyzed the data, authored or reviewed drafts of the article, and approved the final draft.

Yang Ching Chen conceived and designed the experiments, authored or reviewed drafts of the article, and approved the final draft.

The following information was supplied relating to ethical approvals (i.e., approving body and any reference numbers):

Taipei Medical University Institutional Animal Care and Use Committee (IACUC/IACUP) approved all animal procedures.

The following information was supplied regarding data availability:

The data is available at figshare: Chen, Yang Ching (2025). Sleep deprivation and Gut Microbiome–Peer J. figshare. Dataset. https://doi.org/10.6084/m9.figshare.28713239.v1,

The gut microbiome 16s RNA data is available at NCBI: PRJNA1260592 (BioProject), SAMN48399770 (BioSample).

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
