# Peer review of "Chronic sleep deprivation is associated with delayed puberty onset in rats, activation of proinflammatory cytokines and gut dysbiosis"

_PeerJ, doi:10.7717/peerj.19668_

## Round 0.1 · original submission · Major Revisions

Dear authors, I ask you to respond very carefully to all the reviewers' fundamental comments. I hope that the corrections you have made to the manuscript will allow the reviewers to approve the new version of this article.

**Language Note:** The review process has identified that the English language must be improved. PeerJ can provide language editing services - please contact us at [email protected] for pricing (be sure to provide your manuscript number and title). Alternatively, you should make your own arrangements to improve the language quality and provide details in your response letter. – PeerJ Staff

·

Basic reporting

Overall, the paper is well-organized and clearly written. The authors have provided an extensive literature review and numerous references. A variety of figures and tables effectively illustrate the results and serve their purpose well. However, several areas require improvement:
1. Language and Clarity: The English language presentation could be enhanced. Some sentences are confusing and need revision. For instance, the fourth sentence of the abstract and lines 242-243 in the PDF document are unclear.
2. Figures and Tables: There are discrepancies between the figure titles/legends and the corresponding figures in the PDF document (e.g. Figure 2B under title Figure 3). Additionally, the tables in the supplementary materials lack titles or explanatory notes. Each table and figure should include a descriptive title to clarify its purpose and content.
3. Reference for CSD Definition: The statement "Sleep deprivation for 4 weeks was considered as CSD" should be supported with an appropriate reference.

Experimental design

The idea of comparing sleep-deprived rats with control rats is solid. The experimental process and data collection methods appear robust, capturing relevant outcomes and potential factors. Nonetheless, several aspects need attention:
1. Causal Language: The authors use causal language in their conclusions, yet there is no mention of randomization or causal analysis methods. To substantiate causal claims, the authors should detail how juvenile rats were randomly assigned to different groups and ensure baseline covariates are reasonably balanced across groups.
2. Control Group Treatment: The manuscript lacks information on how control rats were treated during the experiment. Were they kept in their home cages, or were they placed in the same tank without water for the same duration? A clear explanation of the control conditions is essential.
3. Multiple Testing Corrections: Given the many statistical tests performed, correction for multiple testing is necessary. The authors should focus on key hypotheses and adjust for multiple comparisons if the specified type I error rate cannot cover that many tests in the paper.

Validity of the findings

The data presented seem credible, but there are concerns about the validity of the conclusions:
Nearly all conclusions about the studied factors are presented as causal (e.g., lines 175-181), but the paper does not provide sufficient details on whether and how a randomized controlled experiment was conducted. This weakens the causal claims. The authors should provide such details. If the study does not meet the criteria for a randomized controlled experiment, conclusions should be framed in a more cautious manner.

Additional comments

None

·

Basic reporting

This manuscript deals with the interesting topic of sleep deprivation and changes in the puberty of a murine model. It has sufficient field background and context provided. It can be published at PeerJ. However, the authors need to share all bioinformatic pipeline details and raw data of PacBio Sequencing in a public repository such as SRA-NCBI or EBI-ENA.

Experimental design

The authors need to describe more profoundly all the details about wet and dry-lab (bioinformatics) experiments.

Validity of the findings

The information in the manuscript does not support the conclusions; the authors need to improve this section of their work.

Additional comments

Detailed comments and suggestions are in the attached file; please revise them, and I will be honored to revise the revised version of this manuscript.

Reviewer 3 ·

Basic reporting

The manuscript could benefit from improvements in clarity, grammar, and consistency of terminology. Several grammatical errors and inconsistencies throughout the text should be addressed. Additionally, some statements lack clarity and require rephrasing for better understanding. Ensuring clear and concise language will enhance the readability and professionalism of the manuscript.

Experimental design

The experimental design has some limitations that could be addressed in future studies. The sample size is relatively small, which may affect the statistical power of the analysis. The authors should consider increasing the sample size in future research to enhance the robustness of the findings.

Validity of the findings

While the study provides valuable insights, the authors should be cautious about overstating their conclusions. The study design is correlational, and drawing causal links between sleep deprivation and delayed puberty may not be fully supported by the data. The discussion should acknowledge the limitations of the study and moderate the language regarding causation. Additionally, providing more detailed information on the methodology, including specific procedures and statistical analyses, would strengthen the validity of the findings.

Additional comments

Chronic sleep deprivation causes delayed puberty onset in rats through activating proinflammatory cytokines and alternating the gut microbiome

The authors explore the link between chronic sleep deprivation (CSD) and its relationship to inflammation and gut health in pubertal rats. The main findings of the article that CSD delays puberty, reduces body weight, and alters the weights of organs. These impacts extend to the immune system where there are elevated levels of inflammatory cytokines in the blood, colon, and reproductive organs. The authors point out correlation between gut bacteria, proinflammatory responses and pubertal maturation. The experimental design was a necessary step forward in understanding the nature of the gut microbiome, the immune system and sleep. The descriptive findings illustrate the changes in a small sample of rats after sleep disruption and will no doubt be important for follow up studies that explore causation. However, the findings of the manuscript are overstated and follow up studies would help establish causation. Although important, the manuscript may not be rigorous enough to be published as a standalone manuscript. This study may be more appropriate as “study 1” in a series of studies within a single manuscript. Substantial enhancements to the description of methodology and statistical analysis is required before recommending this study for publication are required.

Major comments
Introduction
-significant grammatical errors in the introduction make it challenging to read
-the introduction does not flow well, jumping around from one topic to another, a better flow of information that helps the reader understand the necessary topics for this manuscript is necessary
-authors must include a direction when saying “affects” or “alters” as it is not meaningful to understand the authors argument
Methods
-the authors need to explain how sleep disruption was performed in detail, they need to explain how blood was collected (serum, plasma, trunk blood, cardiac perfusion.etc?). How did the authors ensure rats were not sleeping after 23:30? If they did sleep, is this a problem? Why 4 weeks of exposure? Why 15 hours per day? What age were the rats when CSD began?
-It is mentioned that pubertal onset was observed, but it does not explain how. Pubertal observation methods should be as detailed as estrous cycling staging
-How was tissue collection completed?
-If the Bradford method was done, the concentration of BSA (if it was used) should be reported. What were the dilution factors (if applicable) and did the authors run duplicates, triplicates, single samples?
-The method for fecal sample collection must be specified, how did the authors ensure that there was no contamination before, during, and after sample collection?
-What were the comparisons performed in the t test? Were these comparisons decided a priori? Why was an ANOVA with a post-hoc test not performed instead to control for Type 1 error?
-the authors are missing substantial detail regarding ELISA analysis including CV%, what the cutoff was, intraassay coefficients, detection range, specificity. etc
Results
-effect size of the comparisons should be reported in the results along with means and error
-The results section should focus on outlining the specific quantitative findings of the statistical tests rather than simply stating the significance of a comparison
-the low power in the statistical tests may be an issue when performing t tests. How was power assessed, why 6 rats per group? Were the authors able to achieve normality – a necessary precondition for performing a t test – with such a low sample size?
Discussion
-240 the authors explored descriptive effects of CSD in a small sample of rats, they did not explore any interventions and so it is out of the scope of the article to state that treating oxidative stress and gut dysbiosis should occur in future studies
-260-262 since this study is descriptive, how are the authors able to verify causation?
-the authors should consider toning down language related to the causal nature of their study as it is only descriptive



Minor comments
Introduction
-Abstract line 44 “secular” is not appropriate, omit the word
-Authors mention “previous studies have demonstrated…” but do not provide citations – is this appropriate for the abstract? The authors mention “altered pubertal onset” but do not provide a direction – does CSD truly “alter” it to both delay and accelerate pubertal onset?
-“the onset of puberty might via elevating proinflammatory cytokines and alteration of gut microbiome composition” I don’t understand, also “affects” needs a direction
-Line 46-47: the definition of pubertal is neither standard nor correct “Puberty is a developmental phase that determines the end of the growth phase and the 47 beginning of the reproductive phase”. Growth does not end at the beginning of puberty
-46-52 authors switch between “adolescents” and “puberty” without providing a definition of adolescence – a phase which rats do not go through
-52 authors define ROS, but don’t mention it again throughout the manuscript, I’m not sure I understand the connection to the rest of the manuscript
-54, there are many cytokines associated with CSD, why these three?
-57 would it be more accurate to suggest that kisspeptin, and not GnRH, are inhibited by cytokines? Which part of the hypothalamus does this occur within?
-61 exploring the research in greater detail is required: “one study observed that 62 the composition of the gut microbiome are different between pubertal and non-pubertal groups” – there is more than one study on this matter, and they have important nuances
-59 authors mention “some studies have shown” but don’t provide a description of what the study had found, for example “sleep deprivation alters the composition of the 60 gut microbiome” is too vague to meaningful support the authors argument
Methods
-78 “totally 24 rats” should be “a total of 24 rats”
-79 “Sleep deprivation is 80 a highly stressful condition; therefore, the body weights of rats were monitored every other day”, please clarify what is meant by this
-124 do the authors mean “mRNA” or are they looking specifically at RNA?
-166 indent paragraph
Discussion
-238 juvenile rats? Shouldn’t this be pubertal rats?
-242 the authors must be more detailed in the age, sex, and type of experimental design of previous studies
-244 what is meant by “clinical studies”? how does this differ from the previously mentioned study?
-246 “Body weight may be positively correlated with the onset of puberty as being malnourished is related to a delay in the onset of puberty in children (Parent et al. 2003).” Please clarify what is meant by this sentence
-266-267 is there any rationale for not looking at TLR4 instead of LBP, since TLR4 is more closely associated with initiating the proinflammatory response
-302 if pubertal monitoring was done through observation, why would kisspeptin and GnRH analysis be required?

Reviewer 4 ·

Basic reporting

1.1_English language, use of punctuation, and syntax should be improved to ensure clarity of the text and avoid typos. Extended revision of the manuscript is recommended. Examples where text can be improved include:
• Title: alteration instead of alternation
• Abstract: mRNA instead of Mrna; other words are not necessary such as might ([…] we hypothesized that CSD affects the onset of puberty might via elevating proinflammatory cytokines), were (…CSD in juvenile rats for 4 weeks were significantly reduced…), etc; use lower case for Predominant, in some cases rat should be changed into rats.
• Abstract: a sentence about methodology should be added, while the sentence on ‘our results revealed that CSD in juvenile rats for 4 weeks…’ could be eliminated
• Lines 60-63: the use of although and however seems odd
• Line70: correct Materials
• Line 189: likely missing higher than…
• Line 213: likely missing some word
• Line 230: ‘the heat maps were shown the correlation...’
• Line 229: (TABLE S4): replace with lower case
• Discussion: lines 263-266
• Discussion line 243, use of ‘but’
• In general, the discussion would benefit from a reorganization and restructuring of sentences

1.2_ The Introduction does not provide a clear picture on the status of the research in the field: for instance, the authors fail to mention their latest study on sleep deprivation and observed changes in pubertal timing in humans and rats (published in Sleep in 2024) and how the current study is related to the previous one. Are these rats the same of those used in the published paper last year? If not, why did the authors repeat the study? Moreover, I feel that other studies on the effect of sleep insufficiency on sex hormones and gut microbiota could have been cited in the introduction or mentioned in the discussion. Examples (not exhaustive):
Puttawong D, Wejaphikul K, Thonusin C, Dejkhamron P, Chattipakorn N, Chattipakorn SC. Potential Role of Sleep Disturbance in the Development of Early Puberty: Past Clinical Evidence for Future Management. Pediatr Neurol. 2024 Dec;161:117-124. doi: 10.1016/j.pediatrneurol.2024.09.010. Epub 2024 Sep 15. PMID: 39368247. Recent review on the topic sleep and puberty

Siervo GEML, Ogo FM, Staurengo-Ferrari L, Anselmo-Franci JA, Cunha FQ, Cecchini R, Guarnier FA, Verri WA Jr, Fernandes GSA. Sleep restriction during peripuberty unbalances sexual hormones and testicular cytokines in rats. Biol Reprod. 2019 Jan 1;100(1):112-122. doi: 10.1093/biolre/ioy161. PMID: 30010983. For the effects of sleep loss on sex hormones

Bigambo FM, Wang D, Niu Q, Zhang M, Mzava SM, Wang Y, Wang X. The effect of environmental factors on precocious puberty in children: a case-control study. BMC Pediatr. 2023 May 1;23(1):207. doi: 10.1186/s12887-023-04013-1. PMID: 37127587; PMCID: PMC10149633. Night sleep and puberty onset

Gunawan SP, Huang SY, Wang CC, Huynh LBP, Nguyen NN, Hsu SY, Chen YC. Sleep deprivation alters pubertal timing in humans and rats: the role of the gut microbiome. Sleep. 2024 Feb 8;47(2):zsad308. doi: 10.1093/sleep/zsad308. PMID: 38065690.

Zheng LM, Li Y. Modifications in the Composition of the Gut Microbiota in Rats Induced by Chronic Sleep Deprivation: Potential Relation to Mental Disorders. Nat Sci Sleep. 2024 Sep 4;16:1313-1325. doi: 10.2147/NSS.S476691. PMID: 39247907; PMCID: PMC11380879.

Wang Z., Yuan K., Ji Y.-B., Li S.-X., Shi L., Wang Z., Zhou X.-Y., Bao Y.-P., Xie W., Han Y., et al. Alterations of the Gut Microbiota in Response to Total Sleep Deprivation and Recovery Sleep in Rats. Nat. Sci. Sleep. 2022;14:121–133. doi: 10.2147/NSS.S334985.

1.3_Figure1A: add SEM bars for body weight
1.4_Figure3A. Please display individual data points in addition to bars, as done for the other plots
1.5_Line 107: specify abbreviations CAT, SOD, GPx activity. Line 118: LBP. Line 137: missing the – sign in 80C. Line 177: AST. Figure legend Figure4: define LDA
1.6_ Statistics: was normality of data verified?
1.7_Figure5 legend. spearman’s requires capital S

Experimental design

2.1_The authors fail to describe whether their control rats were subjected to the same environmental manipulation of the SD groups and how they controlled for the stress induced by the inverted flowerpot method. Where controls also moved to a water tank with larger platforms? Was corticosterone measured in SD and control groups? If controls stayed in their home cages, I feel that the lack of appropriate controls for SD undermines the conclusion of this study and should be acknowledged as a limitation. Moreover, the influence of different corticosterone levels on pubertal onset, growth, and microbial gut composition should be discussed.
2.2_the authors should provide more information on when the SD occurred in relation to the light:dark cycles or refer to Zeitgeber time for the SD procedure. How where light conditions? When were lights turned ON and OFF?
2.3_ More details are needed on food availability during the SD procedure and on whether food consumption was measured in SD and controls. Could SD procedure have affected food consumption? Could some of their results be explained by reduced food intake in the SD group relative to controls rather than by sleep insufficiency per se. For instance, malnutrition can delay puberty onset (PMID: 7113957) whereas excessive eating can anticipate it (PMID: 25538876). How can the authors tease apart this confounding factor?

Validity of the findings

3.1_The authors need to better clarify the relationship between the current study and their previous work published in Sleep in 2024 and the necessity to replicate the experiments and the findings related to puberty onset and body weight.

3.2_The discussion would benefit from adding a paragraph on why sleep insufficiency has opposing results on puberty onset in humans and rats, and on the potential mechanisms involved.

3.4_ The title and conclusions imply a causal relationship between proinflammatory cytokines, changes in gut microbiota and delayed puberty onset, however a mechanistic causal experiment is missing. Therefore, the title and the conclusions should reflect the associative nature of this link. I recommend changing the title to “Chronic sleep deprivation is associated with delayed puberty onset in rats, activation of proinflammatory cytokines and alteration of the gut microbiome” and tone down any sentence that implies a causal link between the changes in gut microbiota and puberty onset

Additional comments

The work by Gunawan and colleagues focuses on a new and relevant topic which is of potential interest for a broad audience. However, I have 3 main concerns: 1) the authors did not clarify the relation between this work and previous work published in Sleep, which reports similar results regarding the delay in puberty onset and decrease in body weight after sleep deprivation; 2) the lack of proper control for the stress induced by the sleep deprivation technique and the effect of a different environment (with water and no bedding) on faecal bacterial composition; 3) the manuscript needs to be thoroughly revised for grammar, spelling and syntax errors in the use of English.
I recommend major revisions before considering for publication.

---

## Round 0.2 · Major Revisions

Dear Dr. Chen,
I hope that you will make final additions and corrections to the manuscript in accordance with the reviewers' comments. You also need to comply with PeerJ's policy on distributing raw and assembled sequencing data in a public repository. You are using FigShare, but FigShare is not listed in PeerJ's trusted repositories.

·

Basic reporting

The authors have resolved all my previous comments.

Experimental design

Thank you for your revisions. I have a few additional suggestions to improve clarity and consistency:

1. Presentation of Results:
The current phrasing “CSD for 4 weeks caused a decrease in body weight in females (98.69 ± 19.77 g, Figure 1A superior section) and males (145.8 ± 25.69 g, inferior section); in addition, pubertal timing also delayed in both genders (females: 36.33 ± 1.17 days, Figure 1B superior section; males: 45.17 ± 1.60 days, Figure 1B inferior section)” does not clearly convey the magnitude of the effect or the comparison to controls with the values in the parentheses.
I recommend reporting both groups side by side and making it clear what are the values calculated, for example (made up numbers):
“Time-averaged body weight over 4 weeks was lower in SDF vs. CF: females 98.7 ± 19.8 g vs. 122.3 ± 18.5 g, and males 145.8 ± 25.7 g vs. 170.6 ± 22.4 g (Figure 1A). Puberty onset was similarly delayed: females 36.3 ± 1.2 days vs. 34.0 ± 1.0 days, and males 45.2 ± 1.6 days vs. 42.8 ± 1.3 days (Figure 1B).”
This format makes both the absolute and relative differences immediately apparent.


2. Multiple Testing Corrections:
The authors mention that only the puberty-onset hypothesis was pre-specified as primary, with all other analyses being secondary or exploratory. Please consider adding a brief statement in the Methods to that effect, for example:
“We pre-specified puberty-onset timing as our primary hypothesis; all other comparisons are exploratory and unadjusted for multiplicity.”
This will make it clear to readers why no multiple-testing correction was applied.

Validity of the findings

Consistency of Causal Language:
In the Results the authors currently state “CSD for 4 weeks caused a decrease in body weight…,” while in the Conclusions the authors write “Our findings suggest that CSD is associated with delayed puberty onset and altered growth profiles in juvenile rats.”
To maintain consistency—and given the small sample size—I suggest using more measured language like “leading to” “association” in the Results section.

Additional comments

None

·

Basic reporting

The authors have addressed well almost all my comments and suggestions, but they need to share all raw data according to PeerJ policies as stated below:

"If you are reporting an assembled gene/genome/peptide etc. then you must deposit the raw reads and the assembled sequence in the appropriate databases (e.g. NCBI archives)."

Experimental design

The authors have addressed well almost all my comments and suggestions, but they need to share all raw data according to PeerJ policies as stated below:

"If you are reporting an assembled gene/genome/peptide etc. then you must deposit the raw reads and the assembled sequence in the appropriate databases (e.g. NCBI archives)."

Therefore, authors need to add their SRA or ENA numbers to their manuscript for their raw data.

Validity of the findings

The authors have addressed well almost all my comments and suggestions, but they need to share all raw data according to PeerJ policies as stated below:

"If you are reporting an assembled gene/genome/peptide etc. then you must deposit the raw reads and the assembled sequence in the appropriate databases (e.g. NCBI archives)."

Therefore, authors need to add their SRA or ENA numbers to their manuscript for their raw data.

---

## Round 0.3 · accepted · Accept

Dear Dr. Chen, I am pleased to inform you that your article has been accepted for publication in our journal. I hope that you will continue to publish articles on this topic of such high quality.

·

Basic reporting

The authors resolved all my comments

Experimental design

The authors resolved all my comments

Validity of the findings

The authors resolved all my comments

Additional comments

None

Reviewer 3 ·

Basic reporting

No comment

Experimental design

No comment

Validity of the findings

No comment

Additional comments

The authors have responded to all the previous concerns adequately and there is nothing new to suggest for further editing.